# The WHEAT pilot trial—WithHolding Enteral feeds Around packed red cell Transfusion to prevent necrotising enterocolitis in preterm neonates: a multicentre, electronic patient record (EPR), randomised controlled point-of-care pilot trial

Chris Gale,[1] Neena Modi,[1] Sena Jawad,[1] Lucy Culshaw,[2] Jon Dorling,[3] Ursula Bowler,[4] Amanda Forster,[5] Andy King,[4] Jenny McLeish,[6] Louise Linsell,[4] Mark A Turner,[7] Helen Robberts,[8] Kayleigh Stanbury,[4] Tjeerd van Staa,[9] Ed Juszczak[4]

For numbered affiliations see end of article.

**Correspondence to**
Dr Chris Gale;
christopher.gale@imperial.ac.uk

## ABSTRACT

**Introduction** Necrotising enterocolitis (NEC) is a potentially devastating neonatal disease. A temporal association between red cell transfusion and NEC is well described. Observational data suggest that withholding enteral feeds around red cell transfusions may reduce the risk of NEC but this has not been tested in randomised trials; current UK practice varies. Prevention of NEC is a research priority but no appropriately powered trials have addressed this question. The use of a simplified opt-out consent model and embedding trial processes within existing electronic patient record (EPR) systems provide opportunities to increase trial efficiency and recruitment.

**Methods and analysis** We will undertake a randomised, controlled, multicentre, unblinded, pilot trial comparing two care pathways: continuing milk feeds (before, during and after red cell transfusions) and withholding milk feeds (for 4 hours before, during and for 4 hours after red cell transfusions), with infants randomly assigned with equal probability. We will use opt-out consent. A nested qualitative study will explore parent and health professional views. Infants will be eligible if born at <30+0 gestational weeks+days. Primary feasibility outcomes will be rate of recruitment, opt-out, retention, compliance, data completeness and data accuracy; clinical outcomes will include mortality and NEC. The trial will recruit in two neonatal networks in England for 9 months. Data collection will continue until all infants have reached 40+0 corrected gestational weeks or neonatal discharge. Participant identification and recruitment, randomisation and all trial data collection will be embedded within existing neonatal EPR systems (BadgerNet and BadgerEPR); outcome data will be extracted from routinely recorded data held in the National Neonatal Research Database.

**Ethics and dissemination** This study holds Research Ethics Committee approval to use an opt-out approach

### Strengths and limitations of this study

► Necrotising enterocolitis (NEC) is a rare but potentially devastating neonatal disease, occurring predominantly in the most preterm infants. Neonatal trials to-date have not been adequately powered to detect realistic reductions in NEC.

► In this prospective, randomised pilot trial we will evaluate the feasibility of a data-enabled neonatal trial with processes embedded within an existing electronic patient record (EPR) system; accuracy and completeness of trial data will be validated at source.

► In this individually randomised, comparative effectiveness trial we will pilot opt-out consent and explore parent and health professional views of this approach in a nested qualitative study.

► We will evaluate the feasibility of EPR-embedded randomised comparative-effectiveness trials using a simplified opt-out consent for efficient, quicker and less resource burdensome neonatal trials at scale.

to consent. Results will inform future EPR-embedded and data-enabled trials and will be disseminated through conferences, publications and parent-centred information.

**Trial registration number** ISRCTN registry ISRCTN62501859; Pre-results.

## BACKGROUND

Necrotising enterocolitis (NEC) is among the most potentially devastating neonatal diseases and has a mortality of up to 33%, the most severe form (requiring surgery or resulting in death) affects about 5% of infants born at

<30 gestational weeks[1]; survivors are at high risk of long-term health[2] and developmental problems.[3 4] Prevention of NEC has been identified as one of the most important research uncertainties in the field of preterm birth.[5] The pathogenesis of NEC is incompletely understood, however a temporal association between red cell transfusion and the subsequent development of the disease is well described.[6 7] This 'transfusion associated NEC' may be more severe[8] with higher mortality.[9 10] The mechanism thought to underpin this relationship links milk feeds and packed red cell transfusion to NEC through altered mesenteric blood flow and intestinal barrier function; this is supported by animal[11 12] and human studies[13 14 15]. Understanding the link between NEC and blood transfusion is of particular importance given that almost all very preterm babies will have a red cell transfusion and many will receive multiple transfusions[16].

Stopping milk feeds around the time of packed red cell transfusion is currently practised in some neonatal settings to reduce the risk of NEC, putatively by maintaining more physiological intestinal blood flow.[17] This practice has not, however, been tested in an adequately powered randomised trial, and there are physiological reasons why stopping milk feeds in preterm infants may lead to harm. Interrupting enteral feeding prolongs the time taken to reach full milk feeds, which is associated with invasive infection[18], and may paradoxically be associated with an increased risk of NEC [19]. One small, single-centre randomised pilot trial has assessed withholding enteral feeds around red cell transfusion but was underpowered to detect a difference in NEC.[20] A systematic review of observational studies[21] identified seven historical control studies including 7492 preterm infants; these studies were at high risk of bias including regression to the mean and ascertainment bias. Pooled results found an association between withholding feeds in the peritransfusion period and a reduced risk of NEC. The authors concluded that adequately powered randomised controlled trials are needed to confirm these findings.

There is considerable variation in current UK practice in relation to withholding enteral feeds during packed red cell transfusion in preterm infants: a 2011 survey of UK neonatal units (68% response rate) demonstrated that 35% of UK units routinely withheld enteral feeds during packed red cell transfusion.[22]

If withholding enteral feeds around the time of packed red cell transfusion reduces the risk of NEC, then implementing this simple practice will reduce the mortality and long-term complications of NEC. Conversely, if the safety of continued feeding can be demonstrated, this will facilitate increased and consistent feeding with breast-milk, which has well described short-term and long-term benefits.

NEC is rare and occurs at a higher incidence in the most preterm infants and so trials targeting NEC need a large number of very preterm infants, who are themselves rare. As a result, no previous trial has been powered to look at NEC and there is no intervention to prevent NEC

supported by high-quality randomised evidence. Methodologies that have been proposed to improve efficiency and recruitment into randomised trials include the use of simplified opt-out approaches to consent,[23] and embedding trial processes into existing electronic patient record (EPR) systems.[24]

The objectives of this pilot trial are:

1. To determine whether a large multicentre trial addressing the following question is feasible: *among preterm infants (patient), does the practice of withholding enteral feeds around the time of blood transfusion (intervention), compared with continued enteral feeding around the time of blood transfusion (comparator), lead to a reduction in severe necrotising enterocolitis (outcome)?*
2. To determine whether clinical trial processes (identifying participants, randomisation and data collection) can be successfully integrated into existing neonatal EPR systems, and whether trial data can be extracted from routinely recorded clinical data held in the National Neonatal Research Database (NNRD).
3. To determine whether using a simplified opt-out consent process is feasible and acceptable to parents and health professionals.

## METHODS

### Design

The WHEAT trial is a randomised controlled, unblinded, multicentre, pilot trial comparing two care pathways. The primary metrics of feasibility are recruitment, data completeness and data accuracy; clinical outcomes include mortality and NEC. Infants will be randomised with a 1:1 allocation ratio (using permuted blocks of variable size), stratified within neonatal unit by gestational age at birth and infant sex. Trial processes will be embedded within neonatal EPR systems and all outcome data will be extracted from data that are routinely recorded within the existing neonatal EPR systems (BadgerNet and BadgerEPR), and held in the NNRD.

The trial will recruit infants from neonatal units within two neonatal networks in England: Northwest London Neonatal Network and Southern West Midlands Neonatal Operational Delivery Network. Recruitment will be for 9 months (15 October 2018 to 30 June 2019), with data collection continuing for a further 3 months, until all trial infants have finished follow-up at 40+0 corrected gestational weeks or neonatal discharge if sooner.

### Eligibility criteria

Inclusion criteria:
1. Preterm birth at <30+0 gestational weeks+days.
   Exclusion criteria:
1. Parent(s) opted out of trial participation.
2. Packed red cell transfusion with concurrent enteral feeds prior to enrolment. (Infants who have received a packed red cell transfusion while nil-by-mouth are eligible; buccal colostrum will not be counted as enteral feeding.)

3. Infants where enteral feeding is contraindicated in the first 7 days after birth (eg, congenital abnormality).

### Interventions

Both comparator pathways of care are standard in the UK; the WHEAT trial is a pilot comparative effectiveness trial. The two care pathways that will be compared are:

1. Withholding feeds around transfusion: all enteral feeds will be discontinued (the infant will be placed nil by mouth) for a period of 4 hours prior to packed red cell transfusion, during the packed red cell transfusion and until 4 hours post packed red cell transfusion. During this period (~12 hours), hydration and blood glucose will be maintained according to local practice, commonly by provision of parenteral nutrition or intravenous dextrose. Four hours after the red cell transfusion has finished, feeds will be restarted in the manner in which they were being received prior to the decision to transfuse. This duration of withholding feeds will follow the approach used in other trials[20] and observational studies,[21] and identified to be the most acceptable in a survey of UK neonatal units.

2. Continuing feeds around transfusion: enteral feeds will continue to be given prior, during and after the packed red cell transfusion, in the manner in which they were being given prior to the decision to transfuse.

Infants will remain allocated to the same care pathway until 34+6 weeks+days gestational age.

In order to ensure that this pragmatic trial is as generalisable as possible to current practice, blood transfusions will be administered when clinically indicated according to local packed red cell transfusion guidelines. Data will be collected about pretransfusion haemoglobin level for trial participants. Other concomitant care, including speed of increase of enteral feeds and choice of milk, for both the withholding feeds around transfusion pathway and the continuing feeds around transfusion pathway of care will be according to locally defined practice.

### Outcomes

Feasibility outcomes:

1. Recruitment: proportion of infants <30 weeks of gestation admitted whose parents agree to trial involvement and the infant is randomised in the WHEAT trial.
2. Retention: proportion of of recruited infants where outcome data are available up to the end of the follow-up period.
3. Compliance: proportion of recruited infants who correctly received their allocated care pathway around all packed red cell transfusions between randomisation and 34+6 gestational weeks+days.
4. Data completeness: proportion of missing data for each data item reported as a baseline characteristic or an outcome.
5. Data accuracy: proportion of cases where the following data items are correctly recorded when compared with source data (clinical notes or EPR data).

a. Severe NEC: all infants who had a diagnosis of non-severe NEC and a random sample of 25% of infants who did not have a diagnosis of NEC will have their source data verified to ensure that they do not meet the criteria for severe NEC; 25% was selected for pragmatic reasons.
b. Spontaneous intestinal perforation.
c. All-cause mortality.
d. Central line associated blood stream infection.

Clinical outcomes:

All clinical outcomes will be assessed from randomisation to 40+0 weeks of gestation or neonatal unit discharge, whichever occurs first.

1. Severe NEC: histologically or surgically confirmed, or recorded in part 1 the death certificate. These infants will be identified as described in Battersby et al,[25] which will include infants recorded as being transferred for surgery
2. Spontaneous intestinal perforation: histologically or surgically confirmed, or recorded in part 1 the death certificate.
3. All-cause mortality.
4. Total duration of neonatal care in days: including all levels of care (intensive care, high dependency care, special care and ordinary care).
5. Duration of any parenteral nutrition in days.
6. Number of days with a central venous line in situ.
7. Number of central line associated blood stream infections defined according to National Neonatal Audit Programme 2017 definition[26]
8. Growth: change in birth weight and head circumference for gestational age SD score.

### Sample size

There is no predefined sample size for this pilot trial. Recruitment (absolute numbers and the rate) will be a primary outcome for the pilot trial. The estimated recruitment target for the pilot trial is up to 250, based on predicted infant throughput at participating neonatal units and assuming 65%–70% recruitment of eligible infants.

### Data collection

Potential participants will be identified through the existing neonatal EPR systems that are widely used across England, Scotland and Wales; BadgerNet (a clinical summary system) or BadgerEPR (a complete EPR system). Baseline data for all infants admitted to neonatal units in the UK are routinely entered into the EPR *admission summary* as part of normal clinical care. These data are updated in real-time and held securely on BadgerNet and BadgerEPR servers. In participating units, data entered electronically into the *admission summary* will be interrogated by the EPR platform in real time to identify and flag infants meeting the WHEAT trial inclusion criteria. When an infant in a participating unit meets the inclusion criteria, this will result in an electronic reminder appearing on the EPR platform at the participating

 

unit. This 'flag' will inform the health professional that the infant is eligible for the WHEAT trial and link to the parent information leaflet. The EPR system will use data (neonatal unit, gestational age and sex) entered as part of the *admission summary* to stratify infants.

Baseline characteristics and clinical outcomes will be extracted from routinely recorded clinical data held in the NNRD. The NNRD holds data from all infants admitted to National Health Service (NHS) neonatal units in England, Scotland and Wales (~90 000 infants annually). Contributing neonatal units are known as the UK Neonatal Collaborative. Data are extracted from point-of-care neonatal electronic health records completed by health professionals during routine clinical care. A defined data extract, the Neonatal Dataset of ~450 data items,[27] is transmitted quarterly to the Neonatal Data Analysis Unit at Imperial College London and Chelsea and Westminster NHS Foundation Trust where patient episodes across different hospitals are linked and data are cleaned (queries about discrepancies and implausible data configurations are fed back to health professionals and rectified).[28]

### Randomisation
Infants will be randomly assigned to either pathway of care in a 1:1 allocation ratio as per a computer-generated randomisation sequence using permuted blocks of various sizes with stratification as described below. The block sizes will not be disclosed to ensure allocation concealment.

Stratification will be by neonatal unit of enrolment and using the following categories:
1. Gestational age at birth.
   – <28+0 weeks+days
   – 28+0 to 29+6 weeks+days.
2. Infant sex.

Infants who are part of a multiple birth set (twins, triplets or higher order multiples) will be randomised as a set to the same pathway of care following feedback from parent representatives, parent organisations including Bliss and TAMBA (Twins and Multiple Births Association) and research involving parents and adult ex-preterm twins.[29]

### Allocation concealment
Infants will be randomised using an online secure central randomisation system which will be embedded into the existing neonatal EPR systems (BadgerNet and BadgerEPR). Randomisation will occur within the EPR to ensure allocation concealment.

### Blinding
The WHEAT trial will be unblinded as it is not possible to mask the different care pathways.

### Statistical methods
The planned main WHEAT trial will be based on a superiority hypothesis; however, the pilot trial is not powered to detect any differences between the intervention arm (withholding feeds) and the comparator arm (continuing feeds).

Therefore, no formal statistical hypothesis testing will be conducted.

Continuous variables will be summarised using means and SD unless their distributions are skewed, in which case medians, 25th quartiles, 75th quartiles and the range (lowest and highest values) will be presented. Dichotomous variables will be presented as frequencies and percentages. In addition, 95% CIs will be presented for the feasibility outcomes. The recruitment rate will be reported for both arms combined, and retention and compliance rates will be reported separately by treatment arm in addition to both arms combined.

### Changes to the statistical analysis described in the original protocol
The original protocol is available as supplementary data. The following changes to the statistical analysis plan were made prior to completion of data collection:
1. The pilot trial will not be performing any comparative analysis of outcomes between trial arms, or conducting any formal statistical hypothesis testing.
2. The denominator for the recruitment rate will be infants <30 weeks of gestation admitted to recruiting sites; the planned denominator (infants who fulfil all of the eligibility criteria and whose parents have been approached) cannot be used as regulatory approval to use these data was not granted.
3. The opt-out rate of parents whose infants are eligible for the trial will not be reported as regulatory approval to use these data was not granted.
4. Data completeness will be reported for each individual data item and not the proportion of eligible infants for which trial items are complete.
5. A random sample of 25% of infants who did not have a diagnosis of NEC recorded in the EPR system had their source data verified to ensure that they did not meet the criteria for severe NEC.
6. All outcome events, including duration of hospital stay and growth scores, were be measured until neonatal unit discharge or 40+0 weeks of gestation, whichever occurs first.

### Steering committee
An independent Trial Steering Committee (TSC) appointed by the study sponsor and approved by the funder (MRC) will oversee the project. The TSC will consist of an independent chair and at least two other independent members. The Chief Investigator and Clinical Trials Unit Director will also sit on the TSC.

### Data monitoring
A Data Monitoring Committee (DMC) independent of the applicants and of the TSC will review the progress of the trial as agreed and provide advice on the conduct of the trial to the TSC and, via the TSC, to the sponsor. The DMC will act according to its charter, which will be agreed at its first meeting.

## Adverse events

Due to the nature of the patient population, neonates in intensive care, a high incidence of adverse events is foreseeable during their routine care and treatment. Consequently, only those adverse events identified as serious adverse events (SAEs) will be recorded for the trial. Unforeseen SAEs and the SAEs associated with the allocated pathway of care must be reported to the Clinical Trials Unit by a member of site staff within 24 hours of becoming aware of the event. Reporting of SAEs will not use existing EPR systems but will use telephone, fax and email systems.

## Registration

This study is registered in ISRCTN.

## Parent, patient and public involvement

The WHEAT pilot trial addresses one of the most important research uncertainties in preterm birth, as identified by over 500 parents, patients, health professionals and researchers.[5] The WHEAT trial has been developed in partnership with parents; protocol author HR is a parent with experience of preterm birth and protocol author LC represents Bliss, the charity for babies born premature or sick; both HR and Bliss have contributed to trial development from inception. Over 400 parents and patients have contributed to the selection of trial outcomes through the COIN project.[30] Parents and Bliss have been involved in developing the opt-out consent process, how this is communicated, in designing information leaflets and posters. The WHEAT trial has parent representatives on oversight committees to ensure that the trial

## Ethics and dissemination

Because both the care pathways that are being compared are part of standard UK practice, WHEAT is using a simplified model of consent. This means that parents will have the WHEAT trial explained to them and will be asked to 'opt out' if they do not want their infant to be randomised and enrolled in the trial. Parents will be approached shortly after their infant is admitted to the neonatal unit (in most cases within the first 24 hours). There is no upper time limit as to when trial discussions can take place. Parents will be able to opt out of the WHEAT trial at any point. Neonatal health professionals will be prompted within the EPR to explain WHEAT to parents of eligible infants and to provide them with an information leaflet. If parents opt out this will be recorded in the EPR. If parents do not opt out and are happy for their infant to take part in WHEAT, randomisation will occur through the EPR. Enrolment of the infant and the allocation will be notified to the local team through the EPR. Because of the opt-out nature of WHEAT there will not be a signed consent form.

A qualitative exploration of the opt-out consent and recruitment process, and trial procedures will be conducted following the end of recruitment. Qualitative interviews will be undertaken with both parents that consented to the trial and health professionals from the recruiting sites.

Due to the common nature of packed red cell transfusion in the trial population (infants born at <30+0 gestational weeks+days), health professionals will explain the WHEAT trial and opt-out process shortly after birth (in most cases within the first 24 hours). A minority of infants will not receive a packed red cell transfusion during their neonatal unit stay. These will not be included in the main analysis population of clinical outcomes.

Results will be presented at national and international academic conferences and published in peer-reviewed scientific publications. Protocol author HR will work with the neonatal charity Bliss to produce parent-centred information for dissemination through social media, online and to be distributed on neonatal units.

## DISCUSSION

Preventing NEC is a recognised research priority in preterm birth[5]; however, there are no preventative interventions supported by high quality evidence. One key reason is because NEC is a rare condition, therefore any trial seeking to detect a realistic reduction in NEC will require recruitment and randomisation of more preterm infants than ever previously achieved. For example, a trial seeking to detect a 25% relative risk reduction in NEC from a background rate of 6%[26] would need to randomise over 9000 infants to have 90% power to detect such a difference with a two-sided 5% significance level. The largest previous individually randomised trial that included preterm infants was the INIS trial[31] which enrolled 3493 infants. Undertaking neonatal trials on this scale will be challenging; for such large trials to be funded and sustainable, they will need to be more efficient, less burdensome and international in scope. There are successful examples of large simple trials in other specialties that can inform neonatal practice: the TASTE trial[32] demonstrated high efficiency and low burden by integrating trial processes within an existing data capture system, and the TRANSFUSE trial[33] demonstrated very high recruitment rates (>75%) through the use of opt-out models of consent. The WHEAT pilot trial will apply these approaches and measure their feasibility and acceptability in neonatal care. If these methodologies can be successfully applied, they will facilitiate efficient, large, simple trials suitable to address the many clinical uncertainties the plague neonatal care.[34]

The WHEAT pilot trial will determine the feasibility of addressing an important clinical question regarding the optimal approach to feeding preterm infants around the time of red cell transfusions, in preparation for a future definitive trial. Currently, there is insufficient evidence to recommend withholding or continuing milk feeds around red cell transfusion in preterm infants because available physiological and observational data are inconclusive.

## Strength and limitations

The proposed trial has a number of strengths. The robustness of core NNRD data (birth weight, sex, length of stay and death) have been previously demonstrated for research purposes,[1 35 36] this pilot trial will prospectively evaluate their accuracy and completeness for clinical trials. The trial will evaluate the feasibility of recruiting infants across two neonatal networks, including smaller neonatal units that do not traditionally recruit into neonatal randomised trials. Limitations include the unblinded nature of the trial and the use of a potentially subjective primary outcome, NEC. We endeavoured to mitigate against these through use of a previously validated, objective definition for NEC.[1]

## CONCLUSION

Neonatal trials to date have been unable to robustly evaluate strategies to prevent major preterm morbidities, such as optimal feeding around transfusion to prevent NEC, because of the large sample sizes required. This protocol describes a prospective, randomised controlled pilot trial to evaluate trial methodologies aiming to efficiently address such neonatal uncertainties.

### Author affiliations
[1]Neonatal Medicine, School of Public Health, Chelsea and Westminster campus, Imperial College London, London, UK
[2]Bliss – The National Charity for the Newborn, London, UK
[3]Division of Neonatal-Perinatal Medicine, Faculty of Medicine, Dalhousie University, IWK Health Centre, Halifax, Nova Scotia, Canada
[4]Clinical Trials Unit, Nuffield Department of Population Health, University of Oxford, National Perinatal Epidemiology Unit, Oxford, UK
[5]Neonatal Unit, James Cook University Hospital, Middlesbrough, UK
[6]Nuffield Department of Population Health, University of Oxford, National Perinatal Epidemiology Unit, Oxford, UK
[7]Women's and Children's Health, Institute of Translational Medicine, University of Liverpool, Liverpool, UK
[8]Parent of Preterm Twins, Bliss – The National Charity for the Newborn, London, UK
[9]Centre for Health Informatics, Division of Informatics, Imaging and Data Science, School of Health Sciences, Faculty of Biology, Medicine and Health, The University of Manchester, Manchester, UK

**Acknowledgements** We are grateful to all the families who agreed to the inclusion of their baby's data in the NNRD, the health professionals who recorded data and the NDAU team, the members of the study steering group and the members of the NU Neonatal Collaborative representing neonatal units that contribute data to the NNRD.

**Contributors** CG, NM, JD, AF, MAT, HR, TvS and EJ conceived the study. CG, NM, SJ, LC, JD, UB, AF, AK, JM, LL, MAT, HR, KS, TvS and EJ contributed to the planning, conduct and reporting of the study, and writing this manuscript. All authors read and approved the final manuscript. HR is a parent of a preterm twins and LC is a representative of Bliss the charity for babies born premature or sick.

**Funding** The trial is funded through a United Kingdom Medical Research Council (MRC) Clinician Scientist Fellowship awarded to CG.

**Competing interests** None declared.

**Patient consent for publication** Not required.

**Ethics approval** Research Ethics Committee approval was granted on 6 July 2018 by London—Bloomsbury Research Ethics Committee (18/LO/0900).

**Provenance and peer review** Not commissioned; externally peer reviewed.

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
