## [Reviewer comments · BMJ Open]

ARTICLE DETAILS

TITLE (PROVISIONAL)	Study Protocol: The WHEAT pilot trial - WithHolding Enteral feeds Around packed red cell Transfusion to prevent necrotising enterocolitis in preterm neonates: a multi-centre, electronic patient record (EPR), randomised controlled point-of-care pilot trial
AUTHORS	Gale, Chris; Modi, Neena; Jawad, Sena; Culshaw, Lucy; Dorling, Jon; Bowler, Ursula; Forster, Amanda; King, Andy; McLeish, Jenny; Linsell, Louise; Turner, Mark; Robberts, Helen; Stanbury, Kayleigh; van Staa, Tjeerd; Juszczak, Ed

VERSION 1 - REVIEW

REVIEWER	Assoc Prof WARICHA JANJINDAMAI , MD PRINCE OF SONGKLA UNIVERSITY Faculty of Medicine Hat-Yai Songkhla Thailand
REVIEW RETURNED	27-Aug-2019

GENERAL COMMENTS	Thank you very much for giving me the opportunity to review this interesting study protocol. The authors did an excellent introduction, methods and well plan pilot studies. The only comment is "a random sample of 25% of infant who did not have NEC record in EPR system...to ensure that they did not meet the criteria of severe NEC". How came with the number of 25%? . Why did the authors not random sample from the mild or suspected NEC in stead of no NEC?
--

REVIEWER	Amy Keir University of Adelaide and the South Australian Health and Medical Research Institute, Australia
REVIEW RETURNED	02-Sep-2019

GENERAL COMMENTS	Thank you for the opportunity to review this paper. The authors present a well designed and important pilot study protocol in a key area of neonatal transfusion practice. The methodology for this pilot study is appropriate and I have no concerns.
--

	My understanding is that the study is underway and it would be useful to have the study dates (planned or otherwise) included in the protocol. A link or otherwise to the original protocol mentioned in this protocol would also be helpful.
--	---

VERSION 1 – AUTHOR RESPONSE

Thank you to both reviewers for their timely and helpful comments, we have responded below.

Reviewer 1

COMMENT 1: Thank you very much for giving me the opportunity to review this interesting study protocol. The authors did an excellent introduction, methods and well plan pilot studies. The only comment is "a random sample of 25% of infant who did not have NEC record in EPR system...to ensure that they did not meet the criteria of severe NEC". How came with the number of 25%? . Why did the authors not random sample from the mild or suspected NEC instead of no NEC?

RESPONSE: In order to ensure that we do not miss cases of NEC we will check data accuracy in all cases of mild/suspected NEC AND 25% of cases where NEC was not recorded – this is described on page 13. The choice of 25% was a pragmatic one, determined by available resources for source data validation. We have added the following text to explain this in the manuscript on page 13.

“...25% was selected for pragmatic reasons”

Reviewer 2

COMMENT 1: My understanding is that the study is underway and it would be useful to have the study dates (planned or otherwise) included in the protocol. A link or otherwise to the original protocol mentioned in this protocol would also be helpful.

RESPONSE: We have added both dates and a link to the protocol.

On page 11, line 197 we have added “(15th October 2018 – 30th June 2019)”

On page 17, line 347 we have added “The original protocol is available as supplementary data” and have uploaded this with the revision.